# A Systematic Review and Narrative Synthesis of the Relationship between Social Support and Binge Drinking among Adolescents and Emerging Adults

**Eva Villar** [1,*] , **Zeltia Martínez-López** [1] , **M. Emma Mayo** [1] , **Teresa Braña** [2] , **Mauro Rodríguez** [2] **and Carolina Tinajero** [1]

1   Department of Developmental and Educational Psychology, Faculty of Psychology, Universidade de Santiago de Compostela, 15782 Santiago de Compostela, Spain
2   Department of Social Psychology, Basic Psychology and Methodology, Universidade de Santiago de Compostela, 15782 Santiago de Compostela, Spain
*   Correspondence: evavillar.garcia@usc.es

**Abstract:** Binge drinking (BD) is a high-risk pattern of alcohol consumption that is remarkably prevalent among teenagers and emerging adults. This pattern is thought to alter social networks, affecting access to social support (SS), which is considered essential for adjustment during transitional periods and may in turn play a proactive role against risk behaviors. In this review, we aim to synthesize the available data on the relationship between BD and SS in teenagers and emerging adults. Therefore, a search on three electronic databases was conducted (Web of Science, PsycInfo and PubMed). Articles were screened using eligibility criteria in line with the investigation question and the methodological quality of the studies were reported. Data were analyzed using a narrative synthesis approach. Cross-sectional and longitudinal data suggested that SS is associated with the onset, frequency, and intensity of BD; this relation varies with age, gender, and source of support (family or peers). From developmental and socio-cognitive points of view, the following conclusions were reached: (a) effects beyond the detrimental consequences of BD must be considered in order to interpret the data, and (b) social support should be taken into consideration in intervention strategies.

**Keywords:** binge drinking; personal social network; individual social capital; social support; adolescence; emerging adulthood

## 1. Introduction

The consumption of alcohol and other drugs is a major challenge in many countries due to the impact on individual health and well-being [1]. In particular, binge drinking (BD) has received much attention as it is considered a high-risk pattern of alcohol consumption [2,3]. BD is defined as the intake of four or more drinks in women (five or more drinks in men) within approximately two hours, at least once during the previous month, leading to a blood alcohol concentration of at least 0.08 g/dL [4,5].

The currently available data indicate a high prevalence of this drinking pattern, particularly among adolescents (12 to 18 years old) and emerging adults (18 to 25 years old). In a survey conducted in the USA, 34.3% of emerging adults reported BD in the previous month [6]. In European countries, one out of three youngsters aged 15 to 24 years reported frequent BD (at least once a week) in the previous year [7]. Despite BD being traditionally more prevalent among males, recent reports suggest that this gender difference may be fading [8]. In fact, BD is nowadays recognized as a normative rite of passage for adolescents and emerging adults in general [9].

Various consequences of BD in adolescents and emerging adults, both at the individual level and in interpersonal relationships, have been identified [3,10,11], and the BD pattern of alcohol consumption is thought to have more severe outcomes than regular

consumption [12]. Furthermore, an early onset age is considered a risk factor for long term consequences and has been associated with a higher probability of developing a substance use disorder in adulthood [13].

The outcomes most frequently experienced by binge drinkers pertain to the social domain [14–16]. Some issues typically considered within this category include neglected obligations, disruption of family relationships, and regretting something that has been said or done [3,12,17]. As these consequences are considered inherently interactive, they are likely to alter personal social networks, affecting role performance and perceived access to social support (SS) [18] and, in turn, negatively affecting the protective role of SS in regard to unhealthy behavior, particularly alcohol consumption.

### 1.1. Social Support as Derived from Personal Social Networks

Social networks are defined as a set of social relationships in which the members maintain stable interactions [19]. The functionality of these networks (e.g., regulation and control of behavior) will be conditioned by structural (e.g., density, homogeneity) and interactional (e.g., level of integration, frequency of contact or reciprocity) properties, as well as by certain personal characteristics (e.g., gender, age, socioeconomic status) [20–22]. A sub-network of significant others, denominated the personal social network, is usually distinguished, as the presence of this group is expected to enhance health and well-being. Sub-networks involving different types of relationships (e.g., family or friends) have also been independently examined [19].

Individual social capital comprises the set of rules, norms, obligations, relationships of reciprocity and trust, social structures and institutional services that members of a social network can access [23,24] and through which these social networks are expanded [22]. These resources, which vary depending on the relationship (e.g., family, school), enable network members to achieve various communal and personal goals [25] and impact their development, health and well-being [26,27].

SS is one of the main derivations of individual social capital [28]. It has been conceived as the perception or experience of resources provided by others, leading to the sense that one is esteemed, valued and forms part of a social network of mutual assistance [29]. The term thus encompasses two different psychosocial dimensions: a behavioral dimension (supportive actions provided or received) and a cognitive dimension (perceived availability of support) [30]. Perceived SS has been assessed from both a global perspective (in terms of general availability and satisfaction with support) [31], by considering its functionality (i.e., specific provisions, such as emotional, social integration or esteem support) [32] and by taking into account its relational base (regarding different sources, such as family or friends, who provide, or could provide, support) [33]. This theoretical framework is depicted in Figure 1.

### 1.2. Social Support during Adolescence and Emerging Adulthood

The size and composition of personal social network vary throughout the lifetime of an individual and have been considered, from a developmental perspective, in relation to the prominence of certain personal goals and normative life events [19]. Thus, during adolescence and emerging adulthood, the number and diversity of social relationships usually increases; the time spent and the frequency of interactions with the family diminish, while time shared with peers increases [34,35]. During this period, gathering knowledge and information becomes a salient goal, and transitional changes are experienced (physical maturation, establishing autonomy and personal identity, dealing with new responsibilities and demands). All of these changes are potential opportunities to acquire new attributes and abilities, but can also be threats to well-being, making individuals more susceptible to engage in risk behaviors such as BD [36].

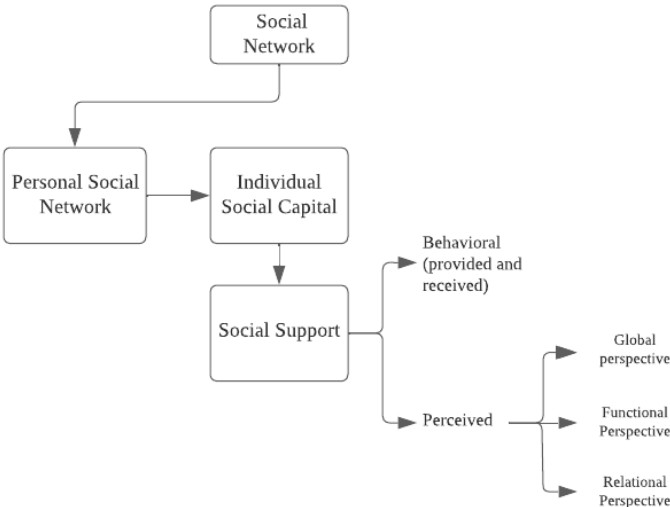

**Figure 1.** Theoretical Relationship Between the Targeted Variables.

SS plays an essential role in adjustment during transitional periods of development. It is thought to act by enhancing one's sense of belonging, self-worth and security, as well as by moderating the appraisal of situations as threatening and also increasing self-confidence to cope with such situations [20,37]. During adolescence and emerging adulthood, a change in the preponderance of the main sources of support is generally observed. The incidence of family support decreases, due to the increased autonomy and independence of young adults [38]. The opposite pattern is observed regarding the relationship with peers, which has greater influence during adolescence and declines towards the end of emerging adulthood [39], mainly through processes of peer selection and socialization [40]. On the other hand, women have been shown to receive and demand more SS than their male counterparts [41].

Social relationships are considered key factors in modulating alcohol consumption during adolescence and emerging adulthood, representing a focal point for research in the field of health [42–45]. In recent years, the protective role of SS has drawn the attention of researchers [46]. Conversely, as previously mentioned, personal social network can also be affected by drinking behavior, suggesting a reciprocal influence between consumption and SS [18]. The present review aims to offer a comprehensive vision of the data and interpretations currently available regarding the bidirectional relationship between SS and BD. Hence, the following objectives were established:

(I) to explore the relationships between SS and BD, and (II) to analyze whether the aforesaid relationships (when identified) vary depending on the type of social bonds (family, peers) and the developmental stage.

## 2. Materials and Methods

This systematic review followed the Preferred Reporting items for Systematic Reviews and Meta-Analyses (PRISMA) guidelines [47]. The Web of Science (WoS), PsycInfo and PubMed databases were consulted, through the advanced search option in the fields "All fields" and/or "Topic" (in "all databases" in the WoS). The original search was performed between November and December 2019, and it was regularly updated through search alerts created in the databases. No time frame was included in the search, in order to capture studies ranging from early theoretical elaborations and empirical research to the most recent contributions. This approach is also consistent with the fact that BD is a recent phenomenon, defined at the beginning of the century [4].

The search expressions used-["Binge Drinking" OR "Heavy Episodic Drinking"] AND "Social Support"-yielded the following number of hits in each database: 244 in the WoS, 144 in PsycInfo, and 132 in PubMed. The initial total was 522, but 289 entries were finally retained after duplicates were removed. The titles and abstracts of the 289 entries

were read, and documents were preselected according to the inclusion/exclusion criteria listed in Table 1. In this regard, the listed exclusion criteria were established in order to stablish the generalizability of our findings, which led us to discard 99 documents. In total, 68 documents were preselected and read in full before being included in or excluded from the final review. Although the focus of the present review was the relationship between SS and BD, studies involving the relationships between both the personal social network and individual social capital and BD were retained, as they may be valuable for interpreting the relationship(s) being explored. No studies on received SS were found. Two team members independently and blindly screened, preselected and then finally selected the documents. Inter-rater reliability, measured using Cohen's Kappa coefficient, was intermediate (k = 0.56) during the preselection stage and high (k = 0.75) during the selection stage. All disagreements during the pre-selection and selection phases were resolved through discussion, and consensus was reached by all authors of the review.

**Table 1.** Eligibility Criteria for the Screening of Documents.

| Inclusion Criteria | Exclusion Criteria |
| --- | --- |
| The BD-SS relation must be addressed in the document. The sample should consist of adolescents (12 to 18 years old) and/or emerging adults (18 to 25 years old). The articles must be written in English, Spanish, French or Portuguese. | Participants with any psychiatric or physical diagnosis. Population under difficult circumstances (violence, pregnancy, COVID-19, etc.). |

The methodological quality was assessed using a template created as a synthesis of the Joanna Briggs Institute Critical Appraisal Tool for Qualitative Research and for Cross-sectional Studies [48,49], as well as the NIH Quality Assessment Tool for Observational Cohort and Cross-sectional Studies [50]. The studies were categorized as intermediate (44%), high (28%) and low quality (28%). The main study limitations were the lack of control of potential confounding variables and failure to obtain representative samples.

Finally, 26 documents (25 empirical studies, 1 critical review [18]) that met the eligibility criteria were selected after detailed, critical screening. After a manual search process, 1 article was added to the list. This process is depicted in a flow diagram (Figure 2).

In the final phase (data extraction), according to the PRISMA guidelines, the main empirical results were summarized in a table. To this end, a matrix was created with the following variables: authors, publication year, quality of the study, measures, and results (see Table 2 in Results section). The possibility of performing a meta-analysis was considered, yet the characteristics of the studies included in the systematic review did not allow us to do it. Therefore, a narrative synthesis was used, to gather the information on the findings and designs of the reviewed studies. Theoretical contributions were also synthetized and integrated in the text.

The whole process was conducted using the Covidence systematic review software (https://www.covidence.org/, accessed on 28 October 2022), the screening and information selection tool selected by Cochrane as the standard production platform for Cochrane Reviews. The complete protocol was registered at PROSPERO (CRD42020184639).

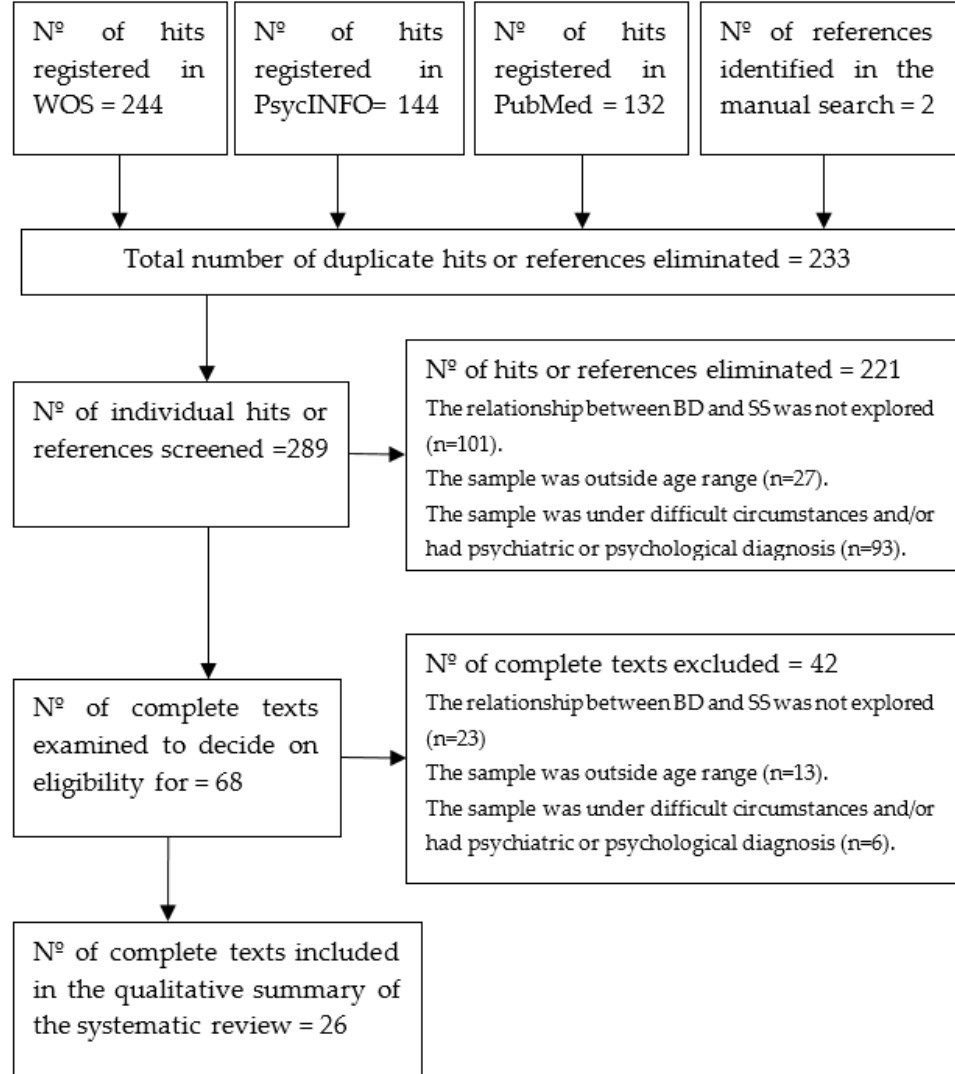

**Figure 2.** Flow Diagram of the Preselection and Selection Phases.

## 3. Results

Here we provide a brief description of the studies analyzed (see details in Table 2). Most (62%) of the documents were published after 2010 and, of these, 25% were published in the last five years (2016-21). Only 11% of the articles were published before 2004 and none were published before 2000, reflecting the recent and increasing interest in the topic. This trend is particularly notable in the US, where most of the studies were performed (55%).

### 3.1. Measures

Regarding how the variables of interest for this review were evaluated, most of the studies (85%) used a single item reflecting both frequency and quantity of drinks or two items (e.g., "On average, how many days per week do you drink alcohol?" and "On a typical day when you drink, on average, how many drinks do you have?") combined in an index in order to identify BD. The AUDIT-C [51] and the Time-Line Follow Back (TLFB) [52] were the standardized measures used to detect BD. The time frame during which BD was evaluated ranged from 1 week and lifetime, with "the last 30 days" being the most frequent interval (30%). Studies that applied a wider time window (e.g., last year, lifetime) mainly focused on adolescence, a developmental stage during which BD is less frequent than in

emerging adulthood. As for the differentiation of cut-off points in males and females, they were only considered in 40% of the studies included in the present review.

Personal social network and individual social capital were measured through diverse indexes, such as network composition and individual popularity.

Finally, among those studies addressing perceived SS, more than half used a variable number of ad hoc items (between 1 and 5) to measure participants' appraisals of global availability of support (e.g., "How much do you feel that your parents care about you"? or "How easy or difficult is it to get help from friends if you need it"?). The Multidimensional Scale of Perceived Social Support (MSPSS) [53] was the most commonly used scale in those studies that used standardized measures. The MSPSS is a 12-item instrument that measures perceived availability of SS from different sources (e.g., family, friends, and significant other). The other instruments used in the different studies identified aspects such as perceived availability of diverse provisions (e.g., emotional, social integration or esteem support) [54–56] and sources [57] of support and/or satisfaction with the support [58,59].

All the studies reviewed focused on the possible protective role of SS against BD. The findings of the various studies are presented below. The studies were generally organized in relation to the type of approach to SS (personal social network, individual social capital and perceived SS), developmental stage and sources of support (teacher, family and peers).

### 3.2. BD and Personal Social Networks

Regarding research on the relationship between BD and personal social network, five studies were identified; all of these reported statistically significant relationships between different network properties and BD both in adolescents and in emerging adults. In particular, in adolescents, the number of groups of friends was associated with a greater likelihood of participation in BD [60], while mainly male composition of personal social network was associated with a higher frequency of BD, especially among boys [61].

During emerging adulthood, belonging to a social network that includes peers considered drinking buddies increases the risk of BD, and a high incidence of BD in the network has been shown to be a significant predictor of individual BD frequency [62]. Furthermore, Lorant & Nicaise (2015) [63] computed the effective size of university students' personal networks as an index of social capital (number of alters minus average number of ties of each contact to other contacts) and found that it negatively predicted BD frequency in emerging adults; the same result was obtained for gender heterophily (i.e., cross-gender relationships), while popularity positively predicted BD.

Special consideration should be given to the research conducted by Hahm et al. (2012) [64], who carried out a longitudinal study in which a sample was tracked from adolescence to emerging adulthood. Low levels of heterogeneity of the personal network (proportion of friends who were not in the same school or grade) and the presence of alcohol-using peers led to a higher frequency of BD during adolescence, although this effect decreased throughout the study period. The opposite trend was observed for popularity, which had an increasingly important impact on BD and was particularly relevant during emerging adulthood.

### 3.3. BD and Individual Social Capital

We identified two articles addressing the relationship between individual social capital and BD. One of these explored the aforementioned relationship in adolescents [65], measuring both trust in people and community participation, but did not obtain significant results. However, Weitzman and Chen (2005) [66] found significant associations between BD and other indexes of individual social capital for emerging adults. Thus, these authors reported that as the level of individual social capital increased (e.g., time spent volunteering), the probability of university students participating in BD decreased. In addition, individual social capital was found to be a significant predictor of the onset of a binge-drinking pattern in first-year students and BD frequency throughout the university period.

*3.4. BD and Perceived Social Support*

Finally, perceived SS was measured globally in three of the studies identified [58,67,68]; none of these reported any significant relationship between perceived SS and BD. Indeed, neither perceived availability nor satisfaction with SS were significant predictors of BD [58,68].

Several studies have specifically explored the relationship between BD and perceived SS from family during adolescence. Five of these [56,69–72] reported a significant and negative relationship, so that high/moderate levels of familiar support were associated with lower probability of and/or frequency of BD.

Of special mention is the study by Wiley (2014) [56], in which perceived parental closeness (i.e., support provision of social integration) was specifically measured and different BD trajectories were established in a sample of adolescents (between 15 and 18 years old). Measurement of the level of social integration differentiated participants who reported infrequent BD (with higher levels of SS) from those who either maintained a constant moderate level of BD or developed a more frequent consumption.

The lack of statistical significance in the relationship between family support and BD observed in the remaining two studies with samples of adolescents [57,59] may be due to the mean age of the study participants (17 and 16.6 years old, respectively), which was higher than in the previously mentioned studies. Indeed, in a study following a sample from adolescence to emerging adulthood, Aseltine and Gore (2000) [69] found an interaction between age and perceived support, in relation to BD frequency.

This pattern is consistent with the results of the five identified studies with samples of emerging adults [55,73–76]. These studies examined the relationship between family support and BD intensity and/or frequency, and the relationship was not found to be statistically significant in any of them.

Regarding perceived SS from peers, four studies reported a positive and significant relationship between this dimension and BD as globally evaluated, in both high school students [69] and university students [77,78]. This relationship can also be recognized in the research regarding specific support provisions, as in the study by Czyzewska & McKenzie (2016) [54], which evaluated emotional support from peers. In a study focused on perceived esteem, Tinajero et al. (2019) [55] noted a curvilineal relationship between this dimension and BD frequency, with an increase in acceptance linked to increasing rates of consumption, followed by a stable level of acceptance at intermediate levels of BD, and a subsequent decline.

**Table 2.** Synthesis of Characteristics and Key Findings of Included Studies.

| Authors | Sample Characteristics | Measures/Intervention | Findings |
|---|---|---|---|
| | | Personal social network | |
| Reifman et al. (2006) [62] [1] | Wave 1/3: 274 first-year university students (63.5% women) from US. Wave 3/3 (1 year later): 43% of the original sample (73.9% women). | Social network: network drinking (1 item), drinking buddies (1 item) BD (2 items): number of binge-drinking episodes (4+/5+ drinks) during the past two weeks. | Percentage of drinking buddies in Wave 1 predicted own alcohol misuse at Wave 2. High average levels of network BD were a significant predictor of individuals' BD |
| Hahm et al. (2012) [64] [2] | 7966 adolescents (11–18 years; 54% women females) from US. Followed for 7 years (3 waves). | Social Network: heterogeneity (2 items), popularity (Bonacich centrality) and density (reciprocity in the nomination). Alcohol using peers (1 item) BD (1 item): frequency of consumption of 5+drinks in a row over the past 12 months. | Lower group heterogeneity and socialization with alcohol using peers were associated with higher frequency of BD at Wave 1. This effect decreased over time. High popularity had a significant effect on the frequency of BD, that grew over time. Density had no association with BD. |

**Table 2.** *Cont.*

| Authors | Sample Characteristics | Measures/Intervention | Findings |
|---|---|---|---|
| Zarzar et al. (2012) [60] [1] | 891 adolescents in Brazil (15–19 years, 59% women). | Social network: "Groups and networks" domain of the Integrated Questionnaire for the Measurement of Social Capital (friendship network characteristics and number of groups of friends) [a]. BD: AUDIT-C (frequency of consumption of 5+ drinks on one occasion) [b]. | Women who valued school friends more than those from hobbies were less likely to report BD. Women who valued school friends less than those from church were less likely to report BD. Female students who reported that the most important group of friends were from school (as opposed to friends from church) had higher odds of BD. Male students with more than 2 groups of friends were more likely to report BD. |
| Lorant & Nicaise (2015) [63] [2] | 478 first-year college students of Psychology (n = 253) and engineering (n = 234) from Belgium. | Social network: popularity, gender-heterophily (Krackhardt E-I index), and heterogeneity. BD (1 item): frequency of consumption of 6+ drinks in the previous year. | High popularity was associated with higher binge-drinking frequency. High levels of network heterogeneity and cross-gender relationships negatively predicted BD. |
| Grard et al. (2018) [61] [2] | 10,932 adolescents (between 14 and 16 years old) from six European cities. | Social Network: Coleman index (same sex/other sex friendships) and school gender balance. BD (1 dichotomous item): frequency of occasions of 5+drinks (at least two) in the last month. | Boys and girls in male-majority schools were more frequent binge drinkers. Other-sex friendship was associated with more BD among girls. |
| | | Individual social capital | |
| Lundborg (2005) [65] [2] | 1346 adolescents (between 12 and 18 years old) from Sweden. | Social capital: trust (1 dichotomous item) and community participation (10 items). BD (1 dichotomous item): intake of 4+ drinks during the past month. | Neither trust nor social participation predicted the probability of BD. |
| Weitzman & Chen (2005) [66] [1] | 27,687 college students (18–24 years, 53.44% women) from US. | Social capital: average daily time committed to volunteering in the past 30 days. BD (1 item): frequency of consumption of 4+/5+ drinks in the past two weeks. | Students with higher social capital showed lower odds of BD. Individual social capital negatively predicted uptake BD among freshman and frequency of BD in college. |
| | | Social Support | |
| Aseltine &Gore (2000) [69] [1] | Wave 1/5: 1208 adolescents (from 14 to 16 years old, 57% women) from US. Wave 5/5 (9 years later): 69% of the original sample (from 22 to 25 years). | Perceived SS from family (3 items) and friends (2 items) from the Perceived Social Support Scale [c]. BD (1 item): frequency of occasions of 5+ drinks on one occasion, during the last week/month. | Statistically significant and negative effect of PSS from parents on BD. This effect decreased over time. PSS from friends was positively associated with BD. |

**Table 2.** *Cont.*

| Authors | Sample Characteristics | Measures/Intervention | Findings |
|---|---|---|---|
| Von Ah et al. (2004) [68] [3] | 161 college students (mean age = 19.7; 73,3% women) from US. | Perceived availability of and satisfaction with SS: Social Support Questionnaire [d]. BD (1 item): frequency of consumption of 5+ drinks in the last 30 days and 6 months. | PSS (availability/satisfaction) did not predict frequency of BD. |
| Windle (2004) [57] [3] | 1049 adolescents (mean age = 17.3, 51.9% women) from US. | Perceived SS from Family: Perceived Social Support Scale [c]. BD (1 item): frequency of consumption of 6+ drinks over the past 6 months | Perceived SS from family did not predict BD. |
| Springer et al. (2006) [71] [2] | 930 adolescents (mean age = 15, 48% women) from El Salvador. | Perceived SS from parents: 7 items based on the Multidimensional Scale of Perceived Social Support [e]. BD (1 dichotomous item): intake of 5+ drinks in the past 30 days. | For women, PSS from parents was a predictor of BD: low levels in this index multiplied by 3 the chances of BD. |
| Ichiyama et al. (2009) [73] [1] | 863 first-year college students (between 18 and 19 years old, 63% women) from the US. 347 included in an intervention program. | Parent Based Intervention designed to increase parental support. BD (1 item): frequency of consumption of 4+/5+ drinks on one occasion, during the last two weeks. | There was no significant intervention effect. |
| Piko & Kovács (2010) [59] [2] | 881 adolescents (mean age = 16.6, 46% women) from Hungary. | Perceived SS from mother and father: Measures of Perceived Social Support [f]. BD (1 item): frequency of consumption of 5+ drinks on one occasion in the past three months. | Perceived parental support did not predict BD. |
| Pearson & Wilkinson (2013) [70] [3] | 13,140 adolescents (mean age = 15.9, 53.1% women) from US. | Perceived SS from family (5 items). BD (1 dichotomous item): intake of 5+ drinks during the previous year. | Perceived family support predicted BD. |
| Schwinn & Schinke (2014) [75] [2] | 400 young adults (mean age = 17.3, 54% women) from US. | Perceived SS from family: Multidimensional Scale of Perceived Social Support [e]. BD (1 item): frequency of consumption of 5+ + drinks on one occasion for the past 30 days. | PSS from family did not predict BD. |
| Zhao (2013) [76] [1] | 140 graduate students (mean age = 27.61, SD=5.76, 73% women) from US. | Perceived SS from family, friends and significant other: Multidimensional Scale of Perceived Social Support [e]. BD (2 items): frequency of consumption of 5+ drinks at least once in the previous year. | PSS was not correlated with BD. PSS was not a predictor of the type of drinker (abstainers, social drinkers, moderate drinkers, binge drinkers, and problematic drinkers). |

**Table 2.** *Cont.*

| Authors | Sample Characteristics | Measures/Intervention | Findings |
|---|---|---|---|
| Wiley (2014) [56] [2] | Wave 1/4: 3341 adolescents (mean age = 15.52) from US. Wave 4/4: 80.3% of the original sample (mean age = 28.52). | Perceived SS (social integration) from parents (4 items) and school (5 items). BD (1 item): frequency of consumption of 5+ in the previous year. | 4 groups were identified considering the onset and progress of BD: (1) increasingly infrequent heavy drinkers, (2) decreasing seldom heavy episodic drinkers, (3) increasing seldom heavy episodic drinkers, (4) increasing occasional heavy episodic drinkers. PSS from parents was a protective factor for groups 3 and 4. PSS from school was a protective factor for groups 2 and 4. For men, PSS from school negatively predicted BD. For women, both PSS from family and school negatively predicted BD. |
| Ryabov (2015) [74] [3] | 1585 young Asian immigrants (from 18 to 26, 51% women) in the US. | Perceived SS from family (4 items). BD (1 item): frequency of consumption of 5+ drinks on one occasion during last year. | Perceived SS from family was not a significant predictor of BD. |
| Stickley et al. (2015) [67] [3] | 3761 men (from 18 to 29 years old) from the former Soviet Union. | Perceived SS (5 items). BD (1 dichotomous item): usually consuming either ≥2 L of beer, ≥750 g of wine, or ≥200 g of strong spirits, on one occasion. | Solitary drinking was more frequent among those participants with low levels of perceived SS. Less than a high level of PSS (low or moderate) was a predictor of solitary drinking. Occasional solitary drinking was a predictor of BD. |
| Czyzewska & McKenzie (2016) [54] [3] | 7476 college students (mean age = 20.72, 65.9% women, 19.5% first generation college student) from US. | Perceived emotional SS from peers (1 item). Perceived need for alcohol use (1 item) BD in the last two weeks (1 item): frequency of consumption of 4+/5+ drinks on one occasion, during the last two weeks. | PSS from friends predicted BD. First generation male students with high levels of SS had higher odds of BD than those with lower levels of SS. |
| Seid (2016) [78] [3] | Adolescents and young adults (from 15 to 29 years) from Denmark. | Perceived SS from friends (1 item). BD (1 dichotomous item): consumption of 5+ drinks on one occasion in the previous 12 months. | Perceived SS from friends was a negative significant predictor of BD. |
| Tinajero et al. (2019) [55] [2] | 484 college students (mean age = 18.25 years; 55,4% women), distributed in 3 groups (control, BD and polyconsuming) from Spain. | Perceived acceptance from friends: Perceived Acceptance Scale [g]. BD: Timeline Followback [h] (days of BD). | Perceived acceptance from the family was higher in the control and BD groups than in the polyconsuming group. Perceived acceptance from friends was higher in BD students than in the control group. A curvilinear relationship between BD and perceived acceptance from friends was identified. |

**Table 2.** *Cont.*

| Authors | Sample Characteristics | Measures/Intervention | Findings |
|---|---|---|---|
| Haardöfer et al. (2020) [58] [1] | Wave 1/6: 3380 college students (between 18 and 25 years old) from the US. Wave 6/6: 2401 college students (between 20 and 27 years old) from the US. | Perceived availability and satisfaction with support: Interpersonal Support Evaluation List [i]. BD (1 item): frequency of consumption of 4+/5+ drinks on one occasion in the past 4 months. | Four trajectories were identified: (1) dabblers, (2) slow decelerators, (3) accelerators, and (4) fast decelerators. Availability of and satisfaction with SS were not significant predictors of BD trajectories. |
| Walsh et al. (2021) [72] [1] | 8221 adolescents (between 11 and 17 years old, 51% female, 18% immigrants) from Israel. | Perceived SS from family: Multidimensional Scale of Perceived Social Support [e]. BD (1 item): frequency of consumption of 5+ drinks in a row in the past 30 days. | Parental support significantly decreased the probability of BD in the non-immigrant sample. |
| Fruehwirth et al. (2021) [77] [2] | Wave 1/2: 1124 college students (mean age = 18.95 years) from the US. Wave 2/2: 474 college students (mean age = 18.9 years) from the US. | Perceived SS from friends: Multidimensional Scale of Perceived Social Support [e]. BD (1 item): frequency of consumption of 4+/5+ drinks in a row in the past 30 days. | Perceived SS from friends significantly increased the probability of BD. |

[a] [79], [b] [51], [c] [80], [d] [81], [e] [53], [f] [82], [g] [83], [h] [52], [i] [84] [1] High quality, [2] Medium quality, [3] Low quality.

## 4. Discussion

The present systematic review attempted to summarize the available evidence and the theoretical arguments regarding the relationship between BD and SS. Since personal social network and individual social capital are considered the basis for SS, studies involving these issues were also reviewed.

Regarding the personal social network, the findings examined suggested that some of the characteristics are associated with greater likelihood and frequency of BD. As previously indicated, Grard et al. (2018) [61] examined the gender composition of the social network in adolescents and found higher probabilities of BD among males and females whose personal social networks was mainly masculine. In interpreting this result, it is worth considering that BD is traditionally a masculine phenomenon [6]. Although recent prevalence rates indicate that women are gradually approaching the level of consumption in males, gender differences persist, as shown in some of the studies analyzed [59,63]. In line with the investigation on risk factors for substance use, Grard et al. (2018) [61] attributed their findings to a higher prevalence of positive attitudes towards alcohol consumption in mainly masculine networks. On their part, Lorant and Nicaise (2015) [63], they pointed out that gender heterophily may imply integrating both male and female norms, making BD less tempting. This interpretation is consistent with the research regarding gender differences in perception of drinking norms [42,85].

The results presented by Zarzar et al. (2012) [60] with high school students are also consistent with the aforementioned interpretation. Thus, these authors found a positive association between the number of groups of friends in the personal social network and BD frequency in male participants, but not in female participants. The very nature of the group relationships in males and females during adolescence may contribute to this gender bias. Males usually have larger personal social network, are more concerned with attributes relevant to status in the peer group, and their personal social network position is strongly linked to acceptance by the peer group [86]. Therefore, males in large social networks are more likely to try to maintain a prominent position by engaging in risky peer-approved behaviors, such as BD. These interpretations, however, should be regarded carefully, as none of the three studies reviewed that took sex into account, considered

different cut-off points for males and females when measuring BD. This could have led to an underestimation of female BD.

Indeed, some of the data collected in the present review indicated that levels of popularity of adolescents among their peers is positively associated with BD frequency and that this effect would increase during emerging adulthood [64]. This upward trend of popularity as a risk factor for substance use may be a consequence, on the one hand, of the experience of transition from high school to university, which establishes the need to manage separation from friends and family, and to become integrated in the university community and share its culture, which includes alcohol use [36]. On the other hand, social network centrality is related to commitment to group norms [87]; thus, we may expect that "central" individuals are more likely to maintain and reproduce prevalent norms of heavy drinking [40].

Another dimension related to the composition of the peer network that seems to affect drinking is its heterogeneity (of the origin of its members). Thus, in a longitudinal study, Hahm et al. (2012) [64] observed that BD rates were higher in adolescents with a wide range of friends outside the academic field. Relationships outside the academic context may be considered to reflect low school engagement and adjustment, which has been shown to maintain a reciprocal association with alcohol use [62,88–90].

Nevertheless, the relationship between heterogeneity and BD seems to be reversed during university transition, as indicated by the findings of the longitudinal studies conducted by Hahm et al. (2012) [64] and Reifman et al. (2006) [62]. This effect may reflect changes in the social climate and adjustment demands. BD is generally normalized on university campuses, even being considered a rite of passage to adulthood [36]. In addition, university peers typically share two types of general drinking motives, namely social (e.g., conformity, peer acceptance, intimacy) and coping motives (e.g., avoiding aversive emotions, alleviation of tension) [91]. Thus, at least for university students, belonging to a heterogeneous peer network could lessen social pressure to participate in BD by providing different perspectives on consumption, especially when the monitoring role of the family network regarding healthy behaviour is hindered by normative increased autonomy and identity exploration [92].

Finally, it has been shown, both in adolescents [64] and in first-year university students [62], that BD frequency is associated with the presence of binge drinkers in the personal social network. These findings are consistent with those reported in previous studies of social influences on risky drinking and may reflect peer modelling [93].

As regards individual indicators of social capital, the results found in this systematic review may indicate different effects of the various facets of individual social capital. Thus, those facets that account for a high level of social implication, such as altruistic behaviour, could help to prevent emerging adults from participating in BD by providing them with different forms of personal exploration and values that would contribute to identity formation [94].

Regarding SS, the research conducted to date with emerging adults does not confirm the association between BD and global PSS. Until more data on this relationship is collected, it seems worth considering the different roles that different sources and provisions of support may play in relation to BD.

Family support seems to help prevent BD, at least during adolescence, as indicated by the consistent negative association between both dimensions in the aforementioned stage [56,69–71,95]. Several arguments presented by the authors of the reviewed articles may help to interpret these results. On the one hand, family relationships are known to exert monitoring and control functions, helping to promote healthy behaviors and to dissuade teenagers from BD. In particular, parental support is assumed to foster bonds to social conventional values, including social norms against substance consumption, lessening the odds of alcohol use [71]. On the other hand, as family support seems to help in the coping process by enhancing one's sense of belonging, self-worth and security [20,37], adolescents would probably resort to family resources in stressful situations rather than relying on BD

as a coping mechanism [70]. The protective effect of parental support seems to be stronger among girls, which is consistent with greater social acceptance of male independence from family [71,96].

The data gathered in the present review suggest that the relationship between family support and BD becomes less significant as individuals reach the end of adolescence and transit to emerging adulthood. Changes in the parent-child relationship during emerging adulthood may undermine the effectiveness and relevance of parental support [59]. On the other hand, the desire for autonomy and independence that characterizes emerging adulthood may also be linked to some detachment from conventional norms regarding alcohol consumption [75]. Accordingly, the attempts of family to continue to exert their former influence may lead to psychological rejection of parental involvement [73], intergenerational conflict between parent and children [74] and tensions regarding autonomy and connectedness [55]. In this context, substance consumption may be perceived as a gateway to adulthood, even though this activity may imply losing family acceptance when family values are transgressed.

The opposite pattern seems to be true regarding support from peers, which has been positively associated with BD probability and frequency, during both adolescence [69] and emerging adulthood [78], in line with research on the effect of peer selection and socialization on BD [97].

Effects beyond the detrimental consequences of BD must be considered in interpreting these findings, as alcohol consumption seems to provide some benefits. Specifically, four reinforcement outcomes have been highlighted: enhancement (e.g., having fun), coping (e.g., forgetting problems), social (e.g., increasing sociability) and conformity (e.g., fitting in with the group) [98]. These positive outcomes may be more relevant to adolescents and emerging adults than the negative consequences, thus increasing the chances of BD in an already favorable environment [85].

The present review highlights SS as an essential variable to be considered to improve our understanding of BD and for proposing intervention strategies to prevent and treat the BD pattern of consumption. In this regard, promoting social integration of adolescents and emerging adults and involving significant support persons during transitional periods (e.g., transition to university) could help decrease the odds of BD. However, some limitations of the studies conducted to date should be noted. First, a large portion of the reviewed research involves cross-sectional designs, which do not enable causal relationships to be established nor the direction of the relationship to be addressed. Aspects related to the generalization of findings must also be considered, as some studies rely on small and/or non-representative samples and over half of the studies were carried out in the US. Finally, self-report measures are prevalent. These measures, although generally reliable, may be subject to response bias. Future research would benefit from distinguishing different provisions of SS and considering the possible modulating role of gender on the relation between SS and BD. Longitudinal studies would also be valuable to further explore changes in the relationship over time, and the reciprocal influence of BD on SS should be explored.

**Author Contributions:** Conceptualization, C.T.; formal analysis, Z.M.-L.; methodology, E.V., Z.M.-L., M.E.M., T.B. and M.R.; investigation, E.V., Z.M.-L., M.E.M., T.B. and M.R.; writing—original draft preparation, E.V.; writing—review and editing, E.V. and M.E.M.; supervision, C.T.; project administration, C.T.; funding acquisition, C.T.; validation, C.T. All authors have read and agreed to the published version of the manuscript.

**Funding:** This research was funded by the Spanish Ministerio de Ciencia e Innovación–Proyectos de Generación de Conocimiento (Award Number: PID2021-126981OB-I00), co-funded by the European Regional Developmental Fund (FEDER) and by Axudas á Consolidación e Estruturación de Unidades de Investigación Competitivas (GRC, ED431C 2022/17).

**Institutional Review Board Statement:** Not applicable.

**Informed Consent Statement:** Not applicable.

**Data Availability Statement:** Not applicable.

**Conflicts of Interest:** The authors declare no conflict of interest.

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
