# Peer review of "A Systematic Review and Narrative Synthesis of the Relationship between Social Support and Binge Drinking among Adolescents and Emerging Adults"

_2673-995X, doi:10.3390/youth2040041_

Round 1

Reviewer 1 Report

the authors present a review on the relationship between social support and binge drinking in adolescence. Overall, the paper is well-written and the literature search and review are competently executed.

I was only slightly puzzled that the authors start on page1/2 explaining that they in fact assume a dynamic, reciprocal relationship between social support and binge drinking. That seems to be a very wise assumption and it should be noted that the literature as presented does not properly account for this hypothesis.

The authors could improve their paper by clearly stating the direction of effects found in the literature. Much of the paper reads like the direction is generally that much SS equals less BD. The fact that in many cases this direction is reversed, is a much-needed information that should be included in the Abstract. The paper may also benefit from a clearer description of the developmental perspective. The latter may simply be a structural issue as the authors have included the information in the text.

Reviewer 2 Report

This is an interesting review covering 25 studies on the association of Binge Drinking (BD) and social support by family and peers in adolescents and young adults. Overall, BD was positively associated with Social Support from Peers and negatively associated with Social Support from family/parents, and strength of association decreased with age. However, not all included studies found these effects. Overall, the search can easily be replicated, the quality rating is important (and should be included in the Table) and the discussion is thoughtful.

I have a few remarks that might be helpful for improving the paper:

Exclusion criteria: it would be interesting to provide a rationale for excluding participants with psychiatric/physical diagnosis and populations under difficult circumstances and to specify the number of studies that were excluded due to these factors.

Included studies: on page 4, you write that you included 24 empirical and 1 theoretical study – in your Table i identified 25 studies with a clear description oft he sample included that in my understanding would all count as empiricl studies. Please specify what you mean by a „theoretical study“

Gender differences: it seems that some studies defined binge drinking as consuming 5+ drinks irrespective of gender while a more common definition includes 5+ form en and 4+ for femals. This might have lead to an underestimation of female BD and should at least be acknowledged in the discussion.

Cultural issues: most studies have been conducted in the US – it would be helpful to discuss if these studies differed from the studies conducted elsewhere, and the cultural generalizability should also be adressed in the discussion.

Discussion: It would be helpful if the authors could elaborate slightly more on the use of social support for improving intervention strategies, given that this is among the main conclusions drawn in the abstract, especially since one study included in the review (Ichiyama et al. 2009) found no effect of a parent based intervention on BD.

Reviewer 3 Report

This is an interesting and well-written piece of work detailing the relationship between sources of support and binge drinking in adolescents. I have some comments/queries which I’m sure will be easily addressed.

1. When was the search carried out? Please include date of search in paper.

2. Methodological quality – beyond details of the tool used to assess methodological quality and proportion of studies categorised as intermediate, high and low quality there didn’t appear to be any specific information on which studies fell into which category. It is important to know for each study their ratings on the tool so that the reader can digest the findings with that in mind.

3. Was there a possibility of meta-analysis? If so, why was one not performed? If not, it would be worth stating why not.

4. Results lacked quantification. For each study, and for each outcome, there should be figures related to effect sizes, mean differences, estimates of robustness, confidence figures, etc. This information would be good to see included in the large table of studies and findings. It also would be helpful to note some of this information in the synthesis of findings.

Round 2

Reviewer 3 Report

Thanks for addressing the comments/queries posed after initial review. I do still have one concern and that is regarding quantification of results. I requested this after noting that it was a systematic review that reported following PRISMA guidelines, and not that it was noted as a narrative review. The PRISMA guidelines clearly state that this is a required component. I understand the narrative nature of the review, but that is not reflective in the title nor the piece. I think the distinction needs to be made or some quantification provided. Thanks

Author Response

In response to Reviewer 3 (Round 2)

Comment: (…) I understand the narrative nature of the review, but that is not reflective in the title nor the piece. I think the distinction needs to be made or some quantification provided.

Response: Attending the Reviewer’s suggestion, we have indicated the narrative nature of the synthesis of gathered data in the tittle, and the abstract and the method sections.
